# Joint Navigation and Manipulation Planning with 3D Interaction Chains

**Keming Zhang**[1][2]  **Sixian Zhang**[1][2] [†] **Xinhang Song**[1][2]  **Hongyu Wang**[1][2]  **Yiyao Wang**[1][2]  **Yingjie Wang**[1]
**Shuqiang Jiang**[2][3]

## Abstract

Open-vocabulary mobile manipulation (OVMM) requires long-horizon navigation in unseen environments and object-centric manipulation. Most existing methods treat navigation and manipulation as separate stages, which can yield navigation endpoints that are poor for manipulation or manipulation-friendly poses that are globally inefficient. To address this, we propose 3D Interaction Chains (3D-IC), a unified framework that couples multi-stage navigation and manipulation planning. 3D-IC maintains a shared 3D feature map for both skills, generates stage-aligned interaction waypoints, and links them into candidate multi-stage chains. A hierarchical policy then scores these chains by jointly considering feasibility (via VLM reasoning over waypoint-centric 3D features) and transition cost, selecting the best trade-off between success and path efficiency. The robot executes the next waypoint and replans as new observations arrive. Experiments in simulation and on a real Stretch 3 robot demonstrate consistent gains in both task success and trajectory efficiency. The code is available at
`https://github.com/kekeZ66/3D-IC`

## 1. Introduction

Navigation and manipulation are two core capabilities for embodied robots to interact with real-world environments. Existing navigation works (Chang et al., 2023; Wang & Soh, 2024) aim to enable a robot to efficiently discover a target and move to its vicinity in unseen environments, leveraging the entire history of observations. In contrast, manipula-

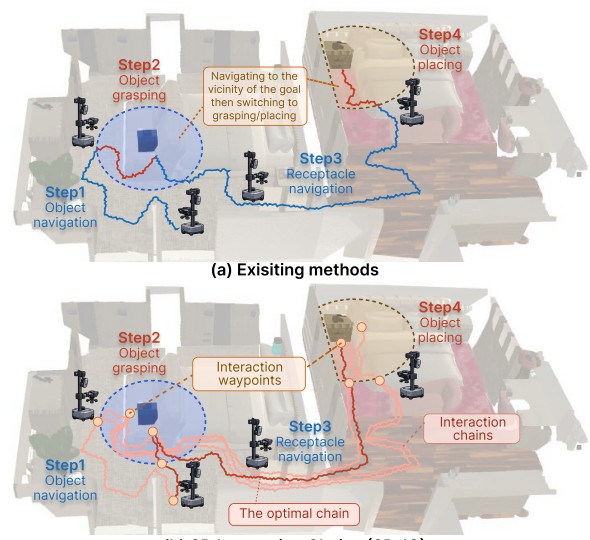

*Figure 1.* (a) Existing methods for OVMM typically plan navigation and manipulation as separate stages, which can result in navigation endpoints that are suboptimal for subsequent interaction. (b) Our 3D-IC jointly plans navigation and manipulation across stages by interaction chains, achieving manipulation-feasible and navigation-efficient plans.

tion (Gu et al., 2023; Chi et al., 2025) focuses on producing temporally continuous action sequences from egocentric observations. However, our work focuses on open-vocabulary mobile manipulation (OVMM) (Yenamandra et al., 2023), which jointly requires long-horizon navigation and dexterous manipulation. Specifically, OVMM is defined as follows: the robot is placed in an unseen environment without access to a global map. Given a user instruction, it must (1) navigate to a specified object, (2) pick it up, (3) navigate to a specified container, and (4) place the object into it. This task alternates between multiple stages of navigation and manipulation, demanding a tight coupling between base navigation and arm manipulation.

Existing OVMM methods can be broadly categorized into modular and reinforcement learning (RL)-based approaches. Modular methods (Yenamandra et al., 2023; Melnik et al., 2023; Liu et al., 2024a) decompose OVMM into separate sub-tasks, each solved by an independent policy. These policies are typically driven by heuristic rules, e.g., exploring frontiers semantically similar to the target (Zhou et al.,

[1]State Key Laboratory of AI Safety, Institute of Computing Technology, Chinese Academy of Sciences, Beijing, China [2]University of Chinese Academy of Sciences, Beijing, China [3]Institute of Computing Technology, Chinese Academy of Sciences, Beijing, China. Correspondence to: Sixian Zhang <sixian.zhang@vipl.ict.ac.cn>.

*Proceedings of the 43rd International Conference on Machine Learning*, Seoul, South Korea. PMLR 306, 2026. Copyright 2026 by the author(s).

2025) or grasping at the object center (Kuzma et al., 2024). However, such independently designed policies and limited heuristics struggle to generalize across diverse situations. RL-based methods instead learn policies through interaction (Gu et al., 2016; Szot et al., 2021; Brohan et al., 2023; Fu et al., 2024), reducing reliance on fixed rules. Yet most existing RL-based methods (Wang et al., 2020; Szot et al., 2021) still use separate policies for navigation and manipulation. This separation can yield navigation endpoints that satisfy "near the object" but are poor for grasping (e.g., blocked by obstacles). Although recent work (Wu et al., 2025b) considers manipulation-aware local motion, it mainly optimizes small-range movement near the target while ignoring cross-stage path planning, producing trajectories that are locally feasible but globally suboptimal, as shown in Fig. 1 (a). Therefore, our motivation is to jointly plan navigation and manipulation across all stages: navigation should arrive at poses that facilitate manipulation, and the full action sequence should be globally efficient over the entire task.

Our goal is joint planning for OVMM, while navigation and manipulation differ substantially in both inputs and outputs: navigation typically conditions on the accumulated history of observations (e.g., a semantic map) and predicts discrete base actions (forward/turn), whereas manipulation relies on egocentric observations and outputs continuous 6-DoF end-effector commands. To bridge this gap, a 3D feature map (Wang et al., 2025b; Wang & Lee, 2025) is constructed to fuse map-level context with egocentric visual information, making the representation compatible with both navigation and manipulation. A hierarchical policy is then adopted, consisting of a high-level policy and stage-specific low-level policies. The shared high-level policy takes the 3D feature map as input and predicts interaction waypoints along with discrete action tokens (e.g., move or grasp), while navigation and manipulation are executed by separate low-level policies that produce their respective actions. Notably, the high-level policy outputs not a single waypoint for the current stage, but an interaction chain spanning multiple OVMM stages.

As shown in Fig. 1 (b), in the planning process of the high-level policy, we first generate candidate interaction waypoints on a 3D feature map, conditioned on the observed target object, the specified container, and frontiers in the map. These waypoints are then organized into four stage-specific sets aligned with the OVMM pipeline. Given the agent's current stage, we iteratively connect waypoints across subsequent stage sets until reaching the next missing/empty set, forming a set of candidate interaction chains. For each chain, the features around each waypoint on the 3D feature map are encoded as visual tokens and fed into a VLM to evaluate navigation and manipulation feasibility, while inter-waypoint distances are computed as transition costs. The high-level policy then selects the optimal chain that best trades off feasibility and path cost, producing a globally optimized multi-stage plan.

In summary, we propose 3D Interaction Chains (3D-IC) for the OVMM task in this paper. Our 3D-IC includes: (1) a 3D feature map that captures information needed for both navigation and manipulation, (2) an interaction chain that enables unified planning across multiple stages, and (3) a hierarchical policy that jointly reasons about navigation and manipulation while outputting their heterogeneous action formats. To execute OVMM, the robot uses 3D-IC to select an optimal interaction chain that balances feasibility and path cost, then executes the next interaction waypoint. As new observations arrive, it repeats this procedure to continually update the plan. We evaluate our 3D-IC in an OVMM simulator and on a real Stretch 3 robot in real-world environments. Experimental results show consistent improvements in both stage-wise success rates and overall path efficiency.

## 2. Related Work

**Object Navigation.** Prior object navigation methods relied on supervised learning (Ramakrishnan et al., 2022; Zhang et al., 2024; Yu et al., 2024) and reinforcement learning (Chaplot et al., 2020; Zhang et al., 2021; 2025a; Zhu et al., 2017; Zhang et al., 2023b; 2022). Recently, large models have enabled zero-shot ObjectNav methods that use them as external priors or planners without task-specific training, which are categorized as LLM-based (Wu et al., 2024; Zhou et al., 2023; Wang et al., 2026) or VLM-based methods (Gadre et al., 2023; Yokoyama et al., 2024; Zhang et al., 2026). While object navigation serves as a critical foundation for downstream tasks such as mobile manipulation, existing methods primarily focus on reaching the proximity of the target (e.g., within a distance threshold). They often overlook the practical requirements of downstream tasks and long-horizon temporal dependencies (Qi et al., 2023; 2024), such as whether the agent's final position is conducive to the manipulation task. Consequently, directly integrating current object navigation methods into downstream pipelines remains challenging and sub-optimal. In contrast, our 3D-IC framework jointly models navigation and manipulation within a unified representation. Employing a chained decision-making process, it produces object navigation policies that are inherently conducive to downstream tasks.

**Mobile Manipulation.** Mobile manipulation frameworks are generally categorized into modular pipelines and end-to-end RL methods. End-to-end approaches (Brohan et al., 2023; Fu et al., 2024; Yan et al., 2025) directly control the whole-body motions of the robot; however, they typically struggle to generalize across long horizons and open vocabularies. Modular frameworks (Yenamandra et al., 2023; Liu et al., 2024a; Melnik et al., 2023) decompose tasks into separate stages. Existing work has performed extensive

and valuable optimizations targeting specific stages, such as gaze control (Gupta et al., 2024), object search (Wang & Soh, 2024; Chang et al., 2023; Zhang et al., 2025b; Wang et al., 2024), placement (Ramrakhya et al., 2024; Wang et al., 2025a), or docking points (Wu et al., 2025b;a). However, these methods often lack joint optimization across the entire pipeline despite their strong individual performance, or are limited to optimizing only stage boundaries (Wu et al., 2025b) without considering longer horizons. This configuration hinders the improvement of global performance and efficiency. To address this, our 3D-IC unifies the various stages of mobile manipulation into Interaction Chains for unified modeling and joint optimization, thereby enhancing both success rates and execution efficiency.

## 3. Preliminaries of Mobile Manipulation

**Task definition.** Formally, our task follows the Open-Vocabulary Mobile Manipulation (OVMM) setting (Yenamandra et al., 2023). A mobile robot is placed in an unseen home environment without access to a pre-built map. Instead, it must explore the environment and predict actions based on online RGB-D observations $I_t$. The robot is required to execute open-vocabulary natural-language instructions $g$ of the form: "Move <object> from <start_receptacle> to <goal_receptacle>". Here, <object> denotes a small, manipulable object (e.g., a toy or a cup), while <receptacle> denotes a large piece of furniture that provides a supporting surface for placement (e.g., a table or a countertop). To complete the task, the robot needs to jointly leverage its mobile base and manipulator arm to perform both navigation (searching for the target object and the relevant receptacles) and manipulation (grasping and placing the object).

Each episode can be decomposed into four sequential stages: (1) Object navigation (stage $s_1$): navigating to the vicinity of the target object; (2) Object grasping (stage $s_2$): predicting a feasible manipulation pose and executing the grasp; (3) Receptacle navigation (stage $s_3$): navigating to the vicinity of the goal receptacle; and (4) Object placing (stage $s_4$): executing the placement via coordinated base and arm motion.

A successful episode is defined as the specified <object> being moved from the <start_receptacle> where it begins the episode to any valid <goal_receptacle>. During evaluation, performance is measured at each of the four stages, and the success rate of the final placing stage is reported as the overall task success rate.

**Existing works.** The above four stages can be categorized into navigation-related stages ($s_1$ and $s_3$) and manipulation-related stages ($s_2$ and $s_4$) formulated as:

$$a_t^n = \pi_n(m_t), \qquad a_t^m = \pi_m(I_t) \qquad (1)$$

where $I_t$ denotes the RGB-D observation at time $t$, and $m_t$ denotes an incrementally constructed 2D semantic map that aggregates all observations $\{I_\tau\}_{\tau=1}^t$ from the beginning of the episode. The navigation policy $\pi_n$ takes $m_t$ as input and outputs base actions $a_t^n \in \{\texttt{Forward}, \texttt{Left}, \texttt{Right}\}$, whereas the manipulation policy $\pi_m$ takes the single-step observation $I_t$ as input and outputs continuous 3D end-effector actions $a_t^m \in \mathbb{R}^6$ (i.e., a 6-DoF pose).

Since the two policies differ in both inputs and action spaces, existing approaches, whether modular pipelines or RL-based methods, typically make decisions with independent policies. A common design in this setting is to use $\pi_n$ to navigate the robot to within a distance threshold of the target (e.g., within $1m$), and then switch to $\pi_m$ to generate manipulation actions. The main difference between modular and RL-based approaches lies in how $\pi_n$ and $\pi_m$ are obtained: heuristic rules versus policies learned via RL. However, a decoupled policy can lead to suboptimal planning, i.e., the navigation endpoint may not be suitable for manipulation, and manipulation-feasible viewpoints may not align with navigation-optimal trajectories. Therefore, our motivation is to perform joint planning that couples navigation and manipulation by discovering an interaction chain that is globally optimal across multiple stages.

## 4. 3D Interaction Chains

### 4.1. Unified Modeling of Multi-stage Interaction

To enable unified planning despite the different inputs and action spaces of navigation and manipulation stages, 3D Interaction Chains (3D-IC) improve both the input representation and the action space. On the input side, a 3D map $\mathcal{M}_t$ is constructed to subsume both the 2D semantic map $m_t$ and the egocentric RGB-D view $I_t$ (i.e., ground-plane projection yields $m_t$, and viewpoint rendering recovers $I_t$). On the action side, the policy is decomposed into: (1) a high-level policy operating on $\mathcal{M}_t$ that outputs a sequence of interaction waypoints and action tokens, and (2) low-level local policies that translate the interaction waypoint into either navigation actions or 6-DoF arm motions.

Formally, given RGB-D observations $I_t$, the robot first builds a 3D feature map $\mathcal{M}_t$. Conditioned on the $\mathcal{M}_t$ and goal $g$ (specifying the target object and receptacle), the high-level planner then predicts an optimal interaction chain:

$$c_t^\star = \pi(\mathcal{C}_t, \mathcal{M}_t, g), \quad c_t^\star = \{(w_k, u_k)\}_{k=1}^K \qquad (2)$$

where $\mathcal{C}_t$ denotes the set of candidate interaction chains and $c_t^\star \in \mathcal{C}_t$ is the optimal chain at time $t$. Each interaction

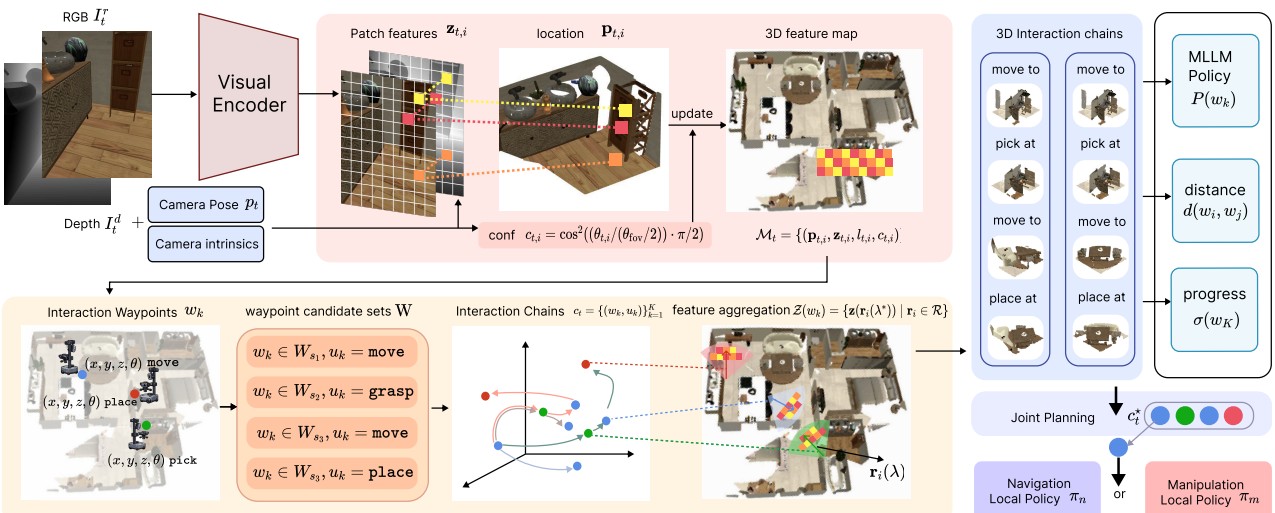

*Figure 2.* **The framework of 3D-IC**. Patch features are extracted from RGB images and projected to establish a 3D feature map. Candidate interaction waypoints are generated on the 3D feature map to construct candidate interaction chains. Subsequently, feature aggregation is performed to obtain $\mathcal{Z}(w_k)$. In the chain decision stage, joint planning is employed to ultimately select the interaction waypoints for execution, which are then dispatched to the local policy based on the task type.

chain is a sequence of interaction waypoints $w_k$ and discrete action tokens $u_k \in \mathcal{U} = \{\texttt{move}, \texttt{grasp}, \texttt{place}\}$. Each waypoint $w_k = (x, y, z, \theta)$ specifies a 3D position $(x, y, z)$ together with the yaw orientation $\theta$ of the robot base. Notably, selecting $c_t^\star$ performs joint planning over multiple stages, which accounts for both long-horizon navigational feasibility and local interaction feasibility. Guided by the high-level plan, the low-level policies execute the next element of the interaction chain:

$$a_t = \begin{cases} \pi_n^l(w_1), & u_1 = \texttt{move} \\ \pi_m^l(w_1, u_1), & u_1 \in \{\texttt{grasp}, \texttt{place}\} \end{cases} \quad (3)$$

where $(w_1, u_1) \in c_t^\star$ denotes the next interaction waypoint in the optimal chain to be executed at time $t$, and $\pi_n^l$, $\pi_m^l$ denote the local navigation and manipulation policies, respectively. In practice, $\pi_n^l$ can be implemented by Fast Marching Method (FMM) (Sethian, 1999), whereas $\pi_m^l$ can be instantiated with off-the-shelf grasping and placing modules (e.g., CuRobo (Sundaralingam et al., 2023), AnyGrasp (Fang et al., 2023)). In contrast to Eq. 1, which employs independent policies for different stages, the proposed 3D-IC unifies multi-stage planning by Eq. 2, while retaining specialized local control via Eq. 3.

### 4.2. 3D-IC Construction

**3D feature map.** As the robot moves through the environment, it continuously acquires egocentric RGB-D observations $I_t = (I_t^r, I_t^d)$ and the camera pose $p_t$, where $I_t^r$ denotes the RGB image and $I_t^d$ denotes the corresponding depth image at time $t$. The 3D feature map $\mathcal{M}_t = \{(\mathbf{p}_{t,i}, \mathbf{z}_{t,i}, l_{t,i}, c_{t,i})\}$ is incrementally constructed from RGB-D observations, where $\mathbf{p}_{t,i} \in \mathbb{R}^3$ denotes a

3D point, $\mathbf{z}_{t,i} \in \mathbb{R}^N$ is a semantic feature vector, $l_{t,i}$ denotes the semantic class label, and $c_{t,i}$ is a confidence score. Concretely, the RGB image is encoded into patch-level semantic features $\{\mathbf{z}_{t,i}\} = \text{ViT}(I_t^r)$ using a Vision Transformer (ViT). In parallel, open-vocabulary segmentation is performed with Grounding DINO (Liu et al., 2024b) and mobile SAM (Zhang et al., 2023a) to assign a semantic label $l_{t,i}$ to each patch. Then, the depth value at the patch center is retrieved from $I_t^d$, and combined with the camera intrinsics and pose $p_t$ to back-project the patch center into 3D, yielding its location $\mathbf{p}_{t,i}$.

Since $\mathcal{M}_t$ is continuously updated, features from multiple viewpoints may be projected to the same 3D location. To enable confidence-aware aggregation, each feature point is associated with a viewpoint-dependent confidence score that captures the reliability of the semantic feature under the current observation. Specifically, using the horizontal field of view (HFOV), the pixel-wise confidence is defined as $c_{t,i} = \cos^2((\theta_{t,i}/(\theta_{\text{fov}}/2)) \cdot \pi/2)$, where $\theta_{t,i}$ is the angular offset between the pixel ray and the optical axis (a larger $\theta_{t,i}$ means the pixel is farther from the image center). $\theta_{\text{fov}}$ denotes the camera's horizontal field of view. The confidence decreases as $\theta_{t,i}$ increases. When multiple features are projected to the same 3D location, the feature with the highest confidence score is retained. The 3D feature map $\mathcal{M}_t$ grounds semantic observations in the 3D geometry of the environment, and subsequent interaction waypoints are selected from it.

In addition, to support collision-free navigation planning, two 2D maps are derived from $\mathcal{M}_t$: an obstacle map $\mathcal{M}_t^o$ and a floor map $\mathcal{M}_t^f$. Specifically, 3D points whose heights are below the agent's traversability threshold are treated as

floor and projected onto the ground plane to form $\mathcal{M}_t^f$, whereas points whose heights exceed the threshold are treated as obstacles and projected to form $\mathcal{M}_t^o$. Furthermore, for efficient exploration when the target has not yet been observed, frontiers are defined as the boundary between floor and unknown areas (i.e., unobserved areas that are neither floor nor obstacle). Following frontier-based exploration (FBE) (Yamauchi, 1997), repeatedly navigating to frontier locations enables the robot to progressively reveal unknown areas and discover targets.

**Interaction waypoint.** To obtain an interaction chain $c_t = \{(w_k, u_k)\}_{k=1}^K$, candidate interaction waypoints $w_k$ and their associated action tokens $u_k$ are first generated from the 3D feature map $\mathcal{M}_t$. Each waypoint is parameterized as $w_k = (x, y, z, \theta)$, consisting of a 3D location $w_k^l = (x, y, z)$ and a yaw orientation $\theta$.

Given the 3D feature map $\mathcal{M}_t$, four stage-specific waypoint candidate sets $W_{s_1}, W_{s_2}, W_{s_3}, W_{s_4}$ are constructed. (1) For the stage-$s_1$ waypoint set $W_{s_1}$, if the target object is observed in $\mathcal{M}_t$, its centroid is estimated from 3D points whose labels match the target class. Then, $K_s$ floor-plane waypoint locations $w_k^l$ are sampled from a ring centered at the centroid with radius $d_n$, and the yaw $\theta$ is set to face the centroid. If the object is unobserved, $W_{s_1}$ is populated with frontier centers in $\mathcal{M}_t$. All $w_k \in W_{s_1}$ are assigned $u_k = \texttt{move}$. (2) For the stage-$s_2$ waypoint set $W_{s_2}$, if the object is observed, $K_s$ candidate locations $w_k^l$ are sampled on the object surface by selecting points within a horizontal slice around the centroid height, with yaw $\theta$ facing the centroid; otherwise, $W_{s_2} = \varnothing$. All $w_k \in W_{s_2}$ are assigned $u_k = \texttt{grasp}$. (3) For the stage-$s_3$ waypoint set $W_{s_3}$, if the target receptacle is observed, $K_s$ floor-layer points are sampled around its centroid at a fixed distance with near-uniform spacing, with yaw facing the centroid; otherwise, frontier centers are used. All $w_k \in W_{s_3}$ are assigned $u_k = \texttt{move}$. (4) For the stage-$s_4$ waypoint set $W_{s_4}$, if the receptacle is observed, the largest supporting surface within the feasible height range is extracted, and $K_s$ candidate placement locations $w_k^l$ are sampled from it, with yaw facing the receptacle centroid; otherwise, $W_{s_4} = \varnothing$. All $w_k \in W_{s_4}$ are assigned $u_k = \texttt{place}$.

**Interaction chain.** The interaction waypoints $\bigcup_{j=1}^4 W_{s_j}$ are further used to construct a set of candidate interaction chains, denoted by $\mathcal{C}_t = \{c_t\}$, where each chain is represented as $c_t = \{(w_k, u_k)\}_{k=1}^K$. The following rules specify how interaction waypoints are linked to form a chain: (1) *Holding-dependent admissibility.* Let $H_t \in \{0, 1\}$ indicate whether the agent is currently holding the target object. If $H_t = 0$, the chain may connect from the current agent pose to waypoints from any stage set; if $H_t = 1$, connections are restricted to receptacle-related waypoints in $W_{s_3} \cup W_{s_4}$. (2) *Monotonic stage progression.* Each waypoint $w$ is assigned

a stage index $\sigma(w) \in \{1, 2, 3, 4\}$ according to the set from which it is drawn. Stage indices are required to be strictly increasing along the chain, i.e., $\sigma(w_{k+1}) > \sigma(w_k)$. Under these rules, a set of candidate interaction chains $\mathcal{C}_t = \{c_t\}$ can be generated at each time step $t$, and updated online as new RGB-D observations are incorporated into $\mathcal{M}_t$.

The waypoint candidate set is updated at each decision step by re-extracting interaction waypoint candidates from the latest map. Accordingly, the interaction chains are reconstructed based on the refreshed waypoint candidates and updated scene observations, from which a set of candidate interaction chains is selected for subsequent decision-making.

### 4.3. Joint Planning with 3D-IC

**Joint planning.** Fig. 2 illustrates the overall construction and decision-making process of 3D-IC. The candidate interaction chains $\mathcal{C}_t = \{c_t\}$ couple waypoints across multiple stages. Joint planning, therefore, amounts to selecting an optimal chain $c_t^\star \in \mathcal{C}_t$. For any pair of waypoints along a chain, the geometric transition cost is defined as the Euclidean distance between their 3D locations, $d(w_i, w_j) = \|w_i^l - w_j^l\|_2$. Beyond geometry, each waypoint is assigned a stage-feasibility probability $P(w) \in [0, 1]$ predicted by a VLM-based policy that queries the local neighborhood features around $w$ in $\mathcal{M}_t$ to judge whether the waypoint can accomplish its corresponding stage (e.g., grasp feasibility for $s_2$ and stable placement for $s_4$). The optimal chain is selected by balancing (1) progress toward completion, (2) waypoint feasibility, and (3) travel cost:

$$c_t^\star = \arg\max_{c_t \in \mathcal{C}} \left[ \sigma(w_K) + \frac{1}{K} \sum_{k=1}^K P(w_k) - \frac{\beta}{|d|} \sum_{k=1}^K d(w_{k-1}, w_k) \right] \tag{4}$$

where $w_0$ denotes the current agent position $p_t$, $|d|$ is a normalization factor, and $\beta > 0$ is a trade-off weight. The term $\sigma(w_K)$ encourages selecting chains whose terminal waypoint corresponds to later stages. During inference, all candidate chains are first pre-ranked according to the progress toward completion and travel cost. The top-$N$ chains are then retained, and the waypoint feasibility term $\sum_{k=1}^K P(w_k)$ is parsed from the VLM output for each retained chain and evaluated using Eq. 4.

To further compute the term $\frac{1}{K} \sum_{k=1}^K P(w_k)$, local features surrounding each waypoint $w_k$ are queried from $\mathcal{M}_t$ and organized via a ray-casting aggregation scheme, which are then provided to a VLM to estimate the feasibility probability. Since a waypoint denotes a target interaction location, sampling features exactly at $w_k$ can be occluded and lack context. To mitigate this, the ray origin $\mathbf{o}$ is placed a fixed distance $\delta$ behind the waypoint along its yaw orientation $\theta$, as $\mathbf{o} = w_k^l - \delta \cdot \mathbf{v}(\theta)$, where $\mathbf{v}(\theta)$ denotes the unit direction vector for angle $\theta$. From $\mathbf{o}$, a bundle of

rays $\mathcal{R}$ is cast, uniformly distributed within a horizontal field of view $\Phi$. Formally, the $i$-th ray is parameterized as $\mathbf{r}_i(\lambda) = \mathbf{o} + \lambda \mathbf{v}(\theta + \phi_i)$, where $\phi_i \in [-\frac{\Phi}{2}, \frac{\Phi}{2}]$ denotes the angular offset. For each ray $\mathbf{r}_i \in \mathcal{R}$, geometry-aware sampling is performed to retrieve the semantic feature at the nearest visible surface. The resulting local semantic representation $\mathcal{Z}(w_k)$ aggregates the features collected along all rays:

$$\mathcal{Z}(w_k) = \{\mathbf{z}(\mathbf{r}_i(\lambda^*)) \mid \mathbf{r}_i \in \mathcal{R}\},$$
$$\text{s.t.} \quad \lambda^* = \underset{\lambda \in [\lambda_{\min}, \lambda_{\max}]}{\operatorname{argmin}} \{\lambda \mid \mathbf{r}_i(\lambda) \in \mathcal{M}_{\text{valid}}\} \quad (5)$$

Here, $\mathcal{M}_{\text{valid}}$ denotes the set of coordinates containing valid 3D features. $\lambda^*$ ensures that each feature corresponds to the first valid geometric intersection, thereby mapping the visible local scene structure into the semantic feature space. For a candidate interaction chain, the set of local representations $\{\mathcal{Z}(w_k)\}_{k=1}^{K}$ is treated as visual tokens and fed into VLM together with a textual prompt to estimate the chain-level feasibility $\frac{1}{K} \sum_{k=1}^{K} P(w_k)$ across multi-stages.

**Policy finetuning.** To enable the VLM to accurately estimate the feasibility probability $P(w_k)$ and reason over candidate interaction chains, we fine-tune Qwen2.5-VL-7B on a curriculum of decision-oriented tasks. To ensure representational consistency, the 3D features in $\mathcal{M}_t$ are extracted using the same visual backbone as the VLM, thereby aligning the queried local representation $\mathcal{Z}(w_k)$ with the model's native embedding space.

The fine-tuning process begins with token interpretation, introduced to adapt the VLM to view-organized visual tokens rather than raw image inputs. The model is trained to describe the 3D geometry from the local feature $\mathcal{Z}(w_k)$, thereby establishing the semantic grounding for $P(w_k)$. Given a waypoint, the annotator VLM generates a concise description, covering (1) semantic role, (2) local 3D geometry, and (3) execution-relevant cues(e.g. reachability, grasp stability, and placement suitability).

To further enhance single-stage decision-making, the model performs token preference learning by comparing candidate waypoints for the same instruction and generating rationales for its selections. Building upon waypoint-level understanding, this stage trains the model to contrastively select better candidates. Given two candidate waypoints from the same instance and stage, the annotator VLM provides concise attributions and scores, encouraging decision-oriented reasoning rather than isolated waypoint descriptions.

To bridge the gap between individual waypoint selection and full interaction chain decision-making, the model is further fine-tuned through trajectory selection, which is most directly aligned with the task. Supervision is obtained by executing candidate chains in simulation, with successful attempts labeled as optimal. For unsuccessful attempts, we annotate concise failure attributions that link the final outcome to specific stage-wise decisions, allowing the model to learn from coupled dependencies across the navigation and manipulation phases. Given multiple candidate trajectories from the same task, the annotator VLM analyzes the simulator feedback for each waypoint along each trajectory and provides corresponding reason and scores for each candidate chain.

For training data collection, we traverse scenes to construct 3D feature maps, from which interaction waypoints, 3D tokens, and valid trajectories are collected. Then we execute actions on sampled waypoints to gather simulator feedback and post-execution RGB images. For navigation, feedback indicates whether the agent moves closer to the goal. For manipulation, we acquire detailed feedback from the simulator, including object-receptacle contact, collisions, and position errors. These physical cues are provided to the annotator VLM to attribute the failure reason (e.g.,the object was placed near the receptacle boundary and eventually slipped because the placement orientation was not aligned with the receptacle edge).

In total, we collect 120K instruction-following training samples across these tasks using in-context distillation from teacher models followed by manual refinement, including 30K samples for token interpretation, 30K for token preference learning, and 60K for trajectory selection. The policy is optimized using a standard autoregressive cross-entropy loss: $\mathcal{L}(\theta) = -\sum_{t=1}^{T} \log p_\theta(x_t \mid \mathbf{x}^{\text{prompt}}, x_{<t})$, where the loss is computed only on the answer tokens $x_t$, while the prompt and visual tokens are masked out from the gradient calculation.

## 5. Experiment

### 5.1. Experimental Setup

**Datasets and Metrics**. As detailed in Sec. 3, we adopt the standard Open-Vocabulary Mobile Manipulation (OVMM) baseline setting (Yenamandra et al., 2023). Success rates (SR) are reported for each stage independently. Notably, the success rate of the final Place stage serves as the overall success rate.

In the OVMM benchmark, the "steps" metric calculates the average number of steps across all episodes where the agent actively terminates, including failed episodes. This metric can lead to biased efficiency comparisons, as models can artificially lower the average step count by "giving up" early. To provide a more robust assessment of execution efficiency, we employ the standard Success weighted by normalized inverse Path Length (SPL) (Anderson et al., 2018) metric $SPL = \frac{1}{N} \sum_{i=1}^{N} S_i \frac{l_i}{\max(p_i, l_i)}$, where $p_i$ denotes the actual path length (quantified by the sequence of actions in the simulator), $l_i$ represents the shortest possible path length,

*Table 1.* **Comparison with other 3D Interaction Point Representations.** "PCD Feat." relies solely on geometric features extracted from point clouds. "PCD Feat. + Label" combines these geometric features with explicit semantic labels. "PCD Rendered" utilizes 2D images rendered directly from colored RGB point clouds. In contrast, "Ours" derives interaction point representations by aggregating features from our proposed 3D feature map, capturing richer semantic context.

| Representations | FindObj. | Pick | FindRec. | Place | Avg. SR |
|---|---|---|---|---|---|
| PCD Feat. | 60.7 | 54.7 | 34.9 | 5.9 | 39.1 |
| PCD Feat. + Label | 62.8 | 56.3 | 36.0 | 5.6 | 40.2 |
| PCD Rendered | 65.1 | 60.1 | 36.8 | 6.8 | 42.2 |
| Ours | **69.1** | **64.0** | **55.6** | **21.9** | **52.7** |

and $S_i$ is a binary indicator representing the success of the $i$-th episode. To determine $l_i$, we utilize the simulator's ground truth to calculate the minimum number of steps required to complete the tasks in each stage via the A* navigation algorithm and optimal manipulation planning.

**Implementation Details**. Our approach is built upon the OVMM heuristic baseline. For the 3D feature map, we leverage the visual encoder from Qwen2.5-VL-7B (Bai et al., 2025) to obtain patch features, and the voxel resolution for projection is set to 0.05m. For feature aggregation, $\delta$ is set to 0.5m, and $\Phi$ is set to 90°. The decision-making component employs a fine-tuned Qwen2.5-VL-7B model. For VLM reasoning, we first pre-rank the candidate chains and retain only 6 chains for reasoning.

For real-world experiments, we use the Hello Robot Stretch 3 as the experimental platform. It features a compliant gripper and a 3-wheel base, is equipped with RGB-D cameras, and supports OVMM task execution.

### 5.2. Evaluation Results

**Ablation on 3D Interaction Point Representations**. Table 1 presents the ablation study on different 3D interaction point representations. Our approach, which utilizes a 3D feature map to construct interaction point representations, achieves the highest performance (Row 4). This is attributed to its efficient encoding of semantic information and spatial relationships, making it suitable for both navigation and manipulation tasks. In contrast, methods relying solely on geometric features extracted by encoders like PointNet++ (Qi et al., 2017) (Row 1), or those augmented with explicit semantic labels (Row 2), yield lower performance. This is primarily due to the lack of rich and implicit semantic context required for complex reasoning. Furthermore, while rendering images directly from RGB point clouds (Row 3) appears intuitive, it underperforms our method. We attribute this performance gap to visual voids and incomplete coverage in regions not fully populated by RGB points. In contrast, our method projects patch-level semantics into the map. Each

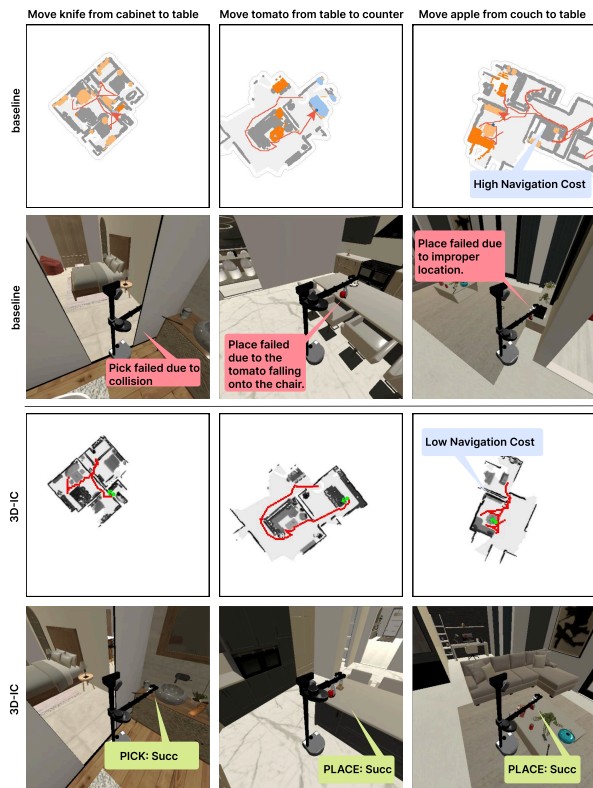

*Figure 3.* Qualitative comparison between the baseline and our method. Each column represents a distinct evaluation case. For each case, Row 1 displays the 2D semantic map from the Heuristic baseline, and Row 2 shows the third-person view at the episode's termination. Row 3 presents the 2D projection of our 3D feature map (where color intensity encodes height), and Row 4 depicts the final third-person view of our method. These examples highlight the advantages of 3D-IC over the baseline, specifically in considering optimal docking orientation, avoiding obstacle occlusion during placement, and generating more efficient execution paths.

feature vector inherently encodes the contextual information of a local region, allowing our representation to maintain semantic continuity and robustness even when geometric coverage is sparse. In summary, our 3D feature map-based interaction point representation effectively captures spatial relationships and structured semantic information.

**Ablation on Chain-based Decision**. Table 2 evaluates the effectiveness of our chain-based decision strategy. Building upon the heuristic baseline (Row 1), employing independent single-stage interaction point decisions at each stage (Row 2) yields significant performance gains. The comparison between Row 2 and Row 1 demonstrates that constructing a unified decision space via interaction points enhances performance across both navigation and manipulation tasks. Furthermore, we extend this to joint decision-making across multiple stages, forming an interaction point chain. Experimental results validate improvements in both success rate and efficiency (SPL). Specifically, jointly optimizing Stages 1 and 2 (Row 3) improves Pick stage performance, likely because the agent selects more optimal grasp points.

*Table 2.* **Ablation study on multi stage planning.** $S_{1,2,3,4}$ denotes performing independent decision-making at each stage. $S_{1-2,3,4}$ and $S_{1,2,3-4}$ involve joint planning for the FindObj-Pick and FindRec-Place phases, respectively. $S_{1-4}$ represents the proposed fully unified decision-making across all four stages.

| | FindObj. | | Pick | | FindRec. | | Place | | Avg. SR |
|---|---|---|---|---|---|---|---|---|---|
| | SR | SPL | SR | SPL | SR | SPL | SR | SPL | |
| Heuristic | 54.1 | 16.7 | 48.5 | 16.6 | 31.3 | 4.8 | 5.1 | 0.8 | 34.8 |
| $S_{1,2,3,4}$ | 66.1 | 23.5 | 62.6 | 23.2 | 54.3 | 10.5 | 15.8 | 2.8 | 49.7 |
| $S_{1-2,3,4}$ | 68.7 | 25.9 | 63.4 | 25.8 | 54.8 | 10.7 | 16.1 | 3.1 | 50.8 |
| $S_{1,2,3-4}$ | 66.3 | 23.6 | 62.5 | 23.4 | 55.2 | 11.8 | 20.7 | 4.5 | 51.2 |
| $S_{1-4}$ | **69.1** | **26.7** | **64.0** | **26.5** | **55.6** | **13.2** | **21.9** | **4.9** | **52.7** |

*Table 3.* **Ablation study on fine-tuning.** TI denotes token interpretation, TP denotes token preference learning and TS denotes trajectory selection.

| TI | TP | TS | FindObj. | Pick | FindRec. | Place | Avg. SR |
|---|---|---|---|---|---|---|---|
| | | | 65.8 | 61.2 | 44.5 | 12.4 | 46.0 |
| ✓ | | | 66.3 | 62.3 | 46.2 | 12.6 | 46.9 |
| | ✓ | | 66.9 | 62.5 | 47.8 | 13.1 | 47.6 |
| | | ✓ | 67.7 | 62.9 | 51.2 | 17.2 | 49.8 |
| ✓ | ✓ | | 67.5 | 62.7 | 50.0 | 15.9 | 49.0 |
| ✓ | | ✓ | 68.3 | 63.2 | 53.8 | 19.8 | 51.3 |
| | ✓ | ✓ | 68.2 | 63.6 | 52.1 | 18.3 | 50.6 |
| ✓ | ✓ | ✓ | **69.1** | **64.0** | **55.6** | **21.9** | **52.7** |

Similarly, joint optimization of Stages 3 and 4 (Row 4) enhances the Place stage performance, which is particularly evident in the selection of suitable placement locations. Finally, when all four stages are integrated into a unified joint decision (Row 5), both performance and efficiency are further maximized. This underscores the critical importance of chain-based decision-making in OVMM, a domain inherently characterized by cross-stage dependencies and heterogeneous task types (i.e., navigation and manipulation).

**Ablation on fine-tuning.** Table 3 evaluates the effectiveness of our three-stage fine-tuning. Comparing rows 3 vs. 5, 4 vs. 6, and 7 vs. 8, we find that adding TI consistently improves performance over training without it. This suggests that TI helps the model better understand the tokenized representation of each waypoint, which in turn benefits the later TP and TS stages. The smaller gain from TI alone (e.g., rows 1 vs. 2) may be because TI mainly improves representation understanding rather than directly optimizing the task-specific objective. Comparing rows 1 vs. 3 and 6 vs. 8 shows that TP also consistently improves performance. This indicates that learning to compare candidate waypoints is beneficial (rows 1 vs. 3) and also helps the subsequent TS stage (rows 6 vs. 8). Finally, comparing rows 4 vs. 7 vs. 8 shows that, although TS is directly aligned with the final task, adding TI and TP still provides further gains over using TS alone. Overall, these results suggest that TI, TP, and TS are complementary and progressively structured: from

*Table 4.* **Real-world Experiment.** "Intra" denotes intra-room tasks where the object and goal receptacle are co-located in the same room. "Cross" refers to cross-room tasks where they are positioned in different rooms, requiring long-horizon navigation.

| Task | Method | FindObj. | Pick | FindRec. | Place | Avg. SR |
|---|---|---|---|---|---|---|
| Intra | Heuristic | 60.0 | 32.0 | 32.0 | 8.0 | 33.0 |
| | 3D-IC | 64.0 | 44.0 | 40.0 | 28.0 | 44.0 |
| Cross | Heuristic | 50.0 | 30.0 | 20.0 | 0.0 | 25.0 |
| | 3D-IC | 70.0 | 40.0 | 30.0 | 20.0 | 40.0 |

understanding a single waypoint, to comparing candidate waypoints, to selecting the best full interaction chain.

**Qualitative Analysis**. Fig. 3 presents a qualitative comparison between 3D-IC and the baseline on the OVMM benchmark. The first column demonstrates that 3D-IC selects a superior docking point during the third stage compared to the baseline. The second column shows that 3D-IC selects the side of the target receptacle (counter) that is unobstructed by the chair, enabling successful placement. Compared to planning on 2D maps, solving navigation in 3D space while jointly considering subsequent manipulation stages ensures that the navigation endpoint better accommodates manipulation requirements (including approach angle and obstacle avoidance). The third column showcases the path efficiency of 3D-IC during joint decision-making, avoiding the excessive path redundancy observed in the baseline. Collectively, these qualitative analyses demonstrate that 3D-IC achieves superior decision-making.

### 5.3. Real-world Evaluation

Table 4 validates the effectiveness of our method in real-world scenarios. We conducted physical robot experiments in a realistic home environment consisting of a living room, a dining area, and two bedrooms. We designed OVMM tasks with two difficulty levels: intra-room tasks, where the object and goal receptacle are located in the same room, and cross-room tasks, where they are positioned in different rooms. For each method, we conducted a total of 35 evaluation episodes, comprising 25 intra-room and 10 cross-room tasks. We deployed both the heuristic baseline and 3D-IC on a Hello Robot platform. Experimental results demonstrate that our method yields significant performance improvements, particularly in long-horizon cross-room tasks and during the Place stage.

Figure 4 illustrates the decision-making and execution process of 3D-IC during a long-horizon cross-room task. After picking up the object, the model successfully determined that the nearest nightstand was unsuitable for placement due to complex surface semantics and high occupancy by other objects. Consequently, the agent navigated back to a nightstand in the initial room to complete the placement, thereby avoiding a potential failure. Note that the colored point

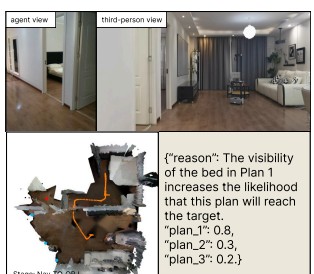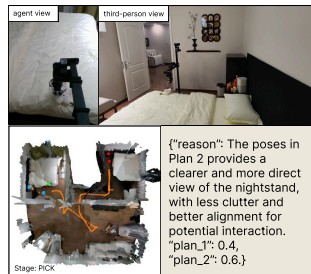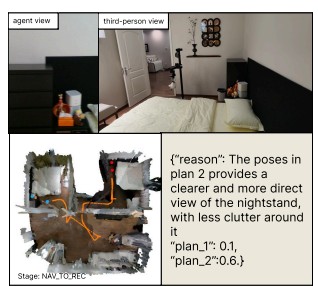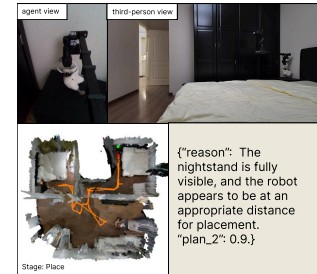

*Figure 4.* Demonstration of real-world experiments and the decision-making process. The top row displays RGB images from the agent's egocentric view and the third-person view of the execution process (captured from different fixed cameras due to the cross-room setting). The bottom-left visualizes the robot's actual trajectory and the selection of interaction waypoints, identifying both candidates (blue) and the selected target (red); note that the colored point cloud is post-processed for visualization purposes only and was not used for policy decision-making. The bottom-right shows the corresponding reasoning content of the planner.

*Table 5.* **Comparisons with the related works**. We report Success Rate (SR) and Success weighted by Path Length (SPL) across all four stages. Note that for Step, lower values (↓) indicate better efficiency, whereas higher values are preferred for SR and SPL. Our method consistently outperforms prior works, establishing new state-of-the-art performance across all metrics.

| Method | FindObj. | | Pick | | FindRec. | | Place(Overall) | | Avg. SR | Step↓ |
|---|---|---|---|---|---|---|---|---|---|---|
| | SR | SPL | SR | SPL | SR | SPL | SR | SPL | | |
| OVMM (Yenamandra et al., 2023) (RL) | 66.6 | 23.9 | 61.1 | 20.9 | 50.9 | 7.0 | 14.8 | 1.4 | 48.4 | 989.7 |
| OVMM (Yenamandra et al., 2023)(Heur.) | 54.1 | 16.7 | 48.5 | 16.6 | 31.3 | 4.8 | 5.1 | 0.8 | 34.8 | 910.3 |
| UniTeam (Melnik et al., 2023) | 66.1 | - | 62.7 | - | 54.7 | - | 18.0 | - | 50.4 | 1027.7 |
| MoManipVLA (Wu et al., 2025b) | 66.1 | - | 62.6 | - | 53.1 | - | 15.8 | - | 49.4 | 1240.5 |
| MoTo (Wu et al., 2025a) | 66.7 | - | 61.0 | - | 49.9 | - | 20.6 | - | 49.6 | 1195.1 |
| 3D-IC(Ours) | **69.1** | **26.7** | **64.0** | **26.5** | **55.6** | **13.2** | **21.9** | **4.9** | **52.7** | **844.1** |

clouds shown in the figure are post-processed for visualization purposes only and were not utilized during the model's decision-making process. Additional real-world execution videos are provided in the supplementary material.

### 5.4. Comparison with SOTA Methods

We compare our method with prior works in Table 5. Our proposed 3D-IC establishes new state-of-the-art performance on the OVMM benchmark. Notably, while our method is built upon the OVMM (Heuristic) baseline, which originally exhibited significantly lower Overall Success Rate (SR) compared to the OVMM (RL) baseline, it surpasses previous methods by a substantial margin.

Furthermore, incorporating the SPL metric into our evaluation provides a more comprehensive assessment of navigation efficiency. The consistently high SPL scores indicate that our method achieves efficient trajectory, rather than merely reducing step counts through premature termination or failure cases (i.e., "giving up early" behaviors).

We hypothesize that the observed performance gains are mainly due to two factors: 1) the unified modeling of multi-stage interactions, which provides a consistent formulation across diverse tasks; and 2) joint construction and planning with 3D-IC, which enables cross-stage decision-making and optimization, thereby reducing the risk of myopic decisions caused by stage-wise decomposition.

### 6. Conclusion

In this paper, we propose 3D Interaction Chains (3D-IC), a framework that unifies the modeling of all task stages in OVMM through joint decision-making to improve both success rates and efficiency. 3D-IC represents both navigation and manipulation tasks uniformly as interaction waypoints within a 3D feature map, enabling joint decision-making across all OVMM stages via interaction chains. We further introduce the SPL metric for the OVMM benchmark to rigorously evaluate the true efficiency of OVMM methods. Experiments on the OVMM benchmark and real-world deployments demonstrate that our 3D-IC method improves success rates while enhancing efficiency, achieving new state-of-the-art performance.

### Acknowledgements

This work was supported by the New Generation Artificial Intelligence-National Science and Technology Major Project (2025ZD0123100).

### Impact Statement

This paper presents work whose goal is to advance the field of Embodied AI and Robotics. There are many potential societal consequences of our work, none of which we feel must be specifically highlighted here.

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

## A. Failure Cases

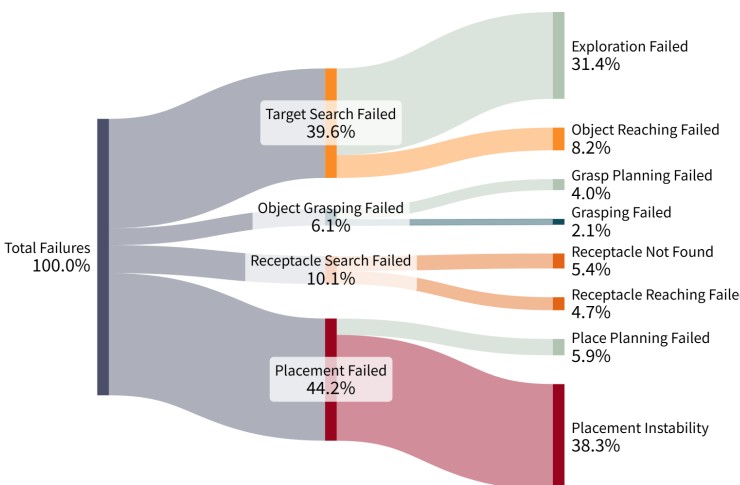

*Figure 5.* Failure cases of 3D-IC on OVMM Dataset. The failure cases are categorized according to the four standard stages of the Open Vocabulary Mobile Manipulation (OVMM) pipeline: Target Search, Object Grasping, Receptacle Search, and Placement. In the navigation phases (Target Search and Receptacle Search), failures are primarily attributed to exploration failures (inability to locate the target or receptacle) and reaching failures. Conversely, in the manipulation phases (Object Grasping and Placement), failures are subdivided into motion planning errors and physical execution errors (specifically, grasping failure and placement instability).

Fig. 5 details the failure distribution across the four task stages, highlighting specific bottlenecks in the 3D-IC framework. Placement Failed emerges as the dominant failure mode (44.2%), driven overwhelmingly by execution-level errors labeled as Placement Instability (38.3%), rather than motion planning failures (5.9%). This suggests that while 3D-IC successfully generates feasible manipulation plans, the final object release remains susceptible to physical dynamics, likely due to suboptimal end-effector stability or contact physics during release.

Target Search Failed is the second most significant error source (39.6%). Decomposing this navigation phase reveals that 3D-IC is primarily limited by Exploration Failed (31.4%)—the inability to semantically locate the target—rather than Object Reaching Failed (8.2%). This indicates that global exploration efficiency and map coverage are more critical constraints for the system than the local reaching policy required to approach the target once detected.

In contrast, the Object Grasping and Receptacle Search stages proved relatively robust under the 3D-IC architecture, contributing only 6.1% and 10.1% to the failure rate, respectively. Notably, grasping failures were split between planning (4.0%) and execution (2.1%), demonstrating the reliability of 3D-IC's manipulation policy compared to the significant challenges faced in the placement and exploration phases.

## B. Prompts

As described in the main text, the VLM evaluates candidate trajectories and generate numerical feasibility scores in a predefined format, from which we parse $\sum_w P(w)$. Listing 1 presents the prompt used for VLM inference. Under this prompt, the VLM is instructed to first assess the feasibility score $P(w)$ for each interaction waypoint, then score each interaction chain accordingly, and finally output the reasoning, together with the score of each chain in Eq. 4.

*Listing 1.* Prompt Template for Inference

```
You are an expert in robot manipulation and navigation. Your task is to analyze the
   feasibility of each candidate plan for successfully completing the given instruction.
   You will provided with the task instruction and some candidate interaction plans,
   where each plan includes a sequence of observations and the specific actions to be
   executed at the corresponding locations.
```

```
The task instruction: <instruction>
Candidate interaction plans: <interaction_chains>

For observations corresponding to the "navigate to <target_object>", consider:
- If the <target_object> is not visible, the likelihood that the surroundings indicate it
    is nearby.
- If the <target_object> is visible, proximity to the object and readiness for grasping.

For observations corresponding to the "pick <target_object>", consider:
- Visibility and clarity of the <target_object> for grasping.
- Accessibility of the object without occlusion or obstruction.
- Suitability of the current angle and position for reliable grasping.

For observations corresponding to the "navigate to <target_receptacle>", consider:
- If the <target_receptacle> is not visible, the likelihood that the surroundings indicate
     it is nearby.
- If the <target_receptacle> is visible, suitability of the current position for placement
    .

For observations corresponding to the "place at <target_receptacle>", consider:
- Obstacles or clutter that could make placement difficult.
- Stability of the object if placed here.
- Suitability of the current angle and position for placement.

Evaluate the feasibility of each interaction waypoint based on the above criteria, and
    then combine these evaluations to determine an overall score for each plan.

Please provide a concise reasoning summary that compares all candidate plans. Then assign
    a raw feasibility score (a continuous value between 0.0 and 1.0) to each plan. Output
    your evaluation strictly in valid JSON format only, with no extra explanation or
    natural language outside the JSON structure.
{
    "reason": "<Concise explanation comparing the plans and justifying the scores>",
    "plan_1": <score_1>,
    "plan_2": <score_2>,
    "plan_3": <score_3>,
    "plan_4": <score_4>,
    "plan_5": <score_5>
    "answer": <plan_ID>,
}
```

To make this process more concrete, an example of the parsing procedure is provided. Listing 2 presents an example input prompt to the VLM, and its corresponding output is shown in Listing 3. Note that the actual input also includes the 3D visual tokens rendered from the 3D feature map. We denote them as <3D tokens> for simplicity in the example.

*Listing 2.* Input Prompt Example for Inference

```
You are an expert in robot manipulation and navigation. Your task is to analyze the
    feasibility of each candidate plan for successfully completing the given instruction.
    You will provided with the task instruction and some candidate interaction plans,
    where each plan includes a sequence of observations and the specific actions to be
    executed at the corresponding locations.

The task instruction: Move tomato from cabinet to counter
Candidate interaction plans:

plan_1: Navigate at <3D tokens>, Pick at <3D tokens>, Navigate at <3D tokens>, Place at <3
    D tokens>.
plan_2: Navigate to <3D tokens>, Pick at <3D tokens>, Navigate at <3D tokens>, Place at <3
    D tokens>.
plan_3: Navigate to <3D tokens>, Pick at <3D tokens>, Navigate at <3D tokens>, Place at <3
    D tokens>.
plan_4: Navigate to <3D tokens>, Pick at <3D tokens>, Navigate at <3D tokens>
plan_5: Navigate to <3D tokens>, Pick at <3D tokens>, Navigate at <3D tokens>
```

```
For observations corresponding to the "navigate to tomato", consider:
- If the tomato is not visible, the likelihood that the surroundings indicate it is nearby
    .
- If the tomato is visible, proximity to the object and readiness for grasping.

For observations corresponding to the "pick tomato", consider:
- Visibility and clarity of the tomato for grasping.
- Accessibility of the object without occlusion or obstruction.
- Suitability of the current angle and position for reliable grasping.

For observations corresponding to the "navigate to counter", consider:
- If the counter is not visible, the likelihood that the surroundings indicate it is
    nearby.
- If the counter is visible, suitability of the current position for placement.

For observations corresponding to the "place at counter", consider:
- Obstacles or clutter that could make placement difficult.
- Stability of the object if placed here.
- Suitability of the current angle and position for placement.

Evaluate the feasibility of each interaction waypoint based on the above criteria, and
    then combine these evaluations to determine an overall score for each plan.

Please provide a concise reasoning summary that compares all candidate plans. Then assign
    a raw feasibility score (a continuous value between 0.0 and 1.0) to each plan. Output
    your evaluation strictly in valid JSON format only, with no extra explanation or
    natural language outside the JSON structure.
{
    "reason": "<Concise explanation comparing the plans and justifying the scores>",
    "plan_1": <score_1>,
    "plan_2": <score_2>,
    "plan_3": <score_3>,
    "plan_4": <score_4>,
    "plan_5": <score_5>
    "answer": <plan_ID>,
}
```

*Listing 3.* Output Example for Inference

```
{
  "reason": "Plan_3 is the most feasible overall, with strong waypoint quality across all
      stages. The poses in Plan_3 provide a grasping position that is closer to the target
       object and free of occlusion, as well as a placement pose that is better aligned
      with the counter. The poses in Plan_1 approaches the counter from its shorter side,
      which is more likely to cause the object to slip or fall. The counter area in Plan_2
       is cluttered, making it less suitable for placement. The scene in Plan_3 looks like
       a bathroom, leading to a low likelihood that a counter is nearby, while the scene
      in Plan_4 appears to be a spacious living room, where finding a counter is still
      possible, but not very likely.",
  "plan_1": 0.6,
  "plan_2": 0.9,
  "plan_3": 0.5,
  "plan_4": 0.1,
  "plan_5": 0.3,
  "answer": "plan_3"
}
```

## C. Runtime Efficiency

The planning phase in our 3D-IC consists of two main steps: (1) generating candidate interaction chains via deterministic spatial operations (avg. 273 ms), and (2) selecting the optimal chain with the VLM (avg. 4,290 ms). This overhead is acceptable for two reasons: (1) 3D-IC planning is performed periodically rather than at every step. (2) real-world physical

execution (e.g., grasping) itself takes some runtime, and our planning runs in parallel. Therefore, the amortized runtime cost per timestep is modest in practice.

