# OpenReview forum: "Joint Navigation and Manipulation Planning with 3D Interaction Chains"
_ICML.cc/2026/Conference — ICML 2026 regular_

### Official Review · Reviewer_jvGm · 2026-02-21

**Soundness:** 2
**Presentation:** 2
**Significance:** 3
**Originality:** 2
**Overall Recommendation:** 4
**Confidence:** 4

**Summary:**

3D-IC tackles the OVMM problem, where a mobile manipulator needs to navigate to a target object, grasp it, go to a target receptacle, and place it. The robot has no prior map and must initially explore the environment.  The primary sensing modality is an RGB-D camera.

The method aims to jointly solve navigation and manipulation, in contrast with previous modular methods that completely decouple both functions. Central to 3D-IC is the 3D feature map  which aggregates visual features across views, weighted by some confidence. The map can be leveraged to ground the relevant objects and sample interaction chains, which are essentially a sequence of (3D waypoint, discrete action) tuples representing the different stages of an OVMM rollout.

A global policy then picks the optimal interaction chain given a cost function that factors in (i) OVMM stage progression, (ii) the waypoint-to-waypoint euclidean distance, and (iii) a VLM-based waypoint feasibility score derived from a local aggregation of the features around the 3D waypoint in the map. The chain is then executed using some standard low-level model-based policies for navigation and grasping. Whenever a target object has not been already observed, the interaction chains are populated with frontier nodes (FBE).

**Compliance With Llm Reviewing Policy:**

Affirmed.

**Final Justification:**

The authors provided a key missing ablation during the rebuttal. While some of the gains are relatively modest, they support all three fine-tuning steps (TI, TP, TS). The discussed changes will also improve the writing. I will update my score to Weak Accept.

**Key Questions For Authors:**

My major concerns are with the **Joint planning and policy finetuning** (see the main review). Answering these questions would improve the experimental design and the soundness of the paper.
1. What is the empirical evidence showing the value of $P(w_k)$ relative to the other terms in Equation 4?
2. What is the empirical evidence justifying the three steps in the fine-tuning procedure described in Appendix C?

My other concerns are related to the presentation.

3. Can the others clarify the generation of interaction chains? Are all admissible chains considered in the selection?
4. Should the specific design of the 3D feature map $\mathcal{M}_t$ be treated as a contribution?

**Limitations:**

The authors explore failure modes and limitations in Appendix A (which should ideally be referred to and summarized in the main paper). In terms of societal impact, the paper does not appear to include an impact statement.

**Strengths And Weaknesses:**

### Strengths
**Motivation.** The method is well motivated. Joint planning over manipulation and navigation is challenging and a promising avenue to significantly improve the performance of mobile manipulation policies and systems.

**Unified input and action spaces.** 3D-IC tackles this problem by unifying the input space through the feature map $\mathcal{M}_t$ and the action space through interaction chains. The feature map is ablated in Table 1.

**Real-world results.** The method has been deployed and tested on a real robot platform. The video shows some interesting rollouts.

### Weaknesses

The listed contributions are (1) the 3D feature map, (2) the interaction chains, and (3) the hierarchical policy. While they are all potentially interesting contributions, I have concerns regarding each contribution.

**3D feature map.** The representation itself reminds me of representations introduced in previous works, such as OpenScene, ConceptFusion, CLIP-Fields, and OK-Robot. The authors should discuss their map representation in the context of this literature. If the contribution results from showing how an existing representation (or a light variation of it) is particularly suitable well for the OVMM task, I would state so explicitly.

**Interaction chains.** The interactions chains are an interesting abstraction for the OVMM problem. As I currently understand the paper, there is (1) a sampling stage to generate interaction waypoints and connect them to yield chains, and (2) a policy to discriminate the optimal chain. The current description of (1) is somewhat vague. Lines 249-265 give some rules but the authors should clarify if all the chains meeting those criteria are considered in the optimization. This would potentially include a lot of candidates unless the waypoint candidate sets are very small. I would also consider describing how the set is updated online.

**Joint planning and policy finetuning.** The optimal chain is chosen by optimizing Equation 4. The VLM-based feasibility scores $P(w_k)$ raise a few concerns.
1. The presentation lacks details. We learn in lines 275-295 (left column) that the fine-tuning process is a complex 3-step procedure involving token interpretation, token preference learning, and trajectory selection using simulation trajectories for supervision. The description in the main paper is hard to follow and the steps are only clarified in appendix, to which no references are given in the main text. I would also clarify exactly how the $P(w_k)$ are retrieved from the VLM.
2. The overall fine-tuning is ablated in Table 5. In my view, some key ablations are missing. First, the entire $P(w_k)$ should be ablated. This would confirm the actual contribution of the VLM over the other terms in Equation 4. Stage progress and waypoint distances likely provide a valuable signal on their own. Second, given the complexity of the fine-tuning procedure, a systematic ablation of each step appears necessary.

**SPL.** The paper should cite [1] for the SPL metric. The final paragraphs also frame using the SPL metric for OVMM as a contribution. While appropriate, I would say SPL is already pretty standard in embodied settings.

[1] Anderson, Peter, et al. "On evaluation of embodied navigation agents." arXiv preprint arXiv:1807.06757 (2018).

### Minor Questions
1. Why consider the Euclidean distance in Equation 4 when the actual shortest path distance could be derived from $\mathcal{M}^f_t$ for navigation steps?
2. Sharing a single representation for navigation and manipulation is appealing. However, during actual robot deployment, localization and mapping errors can degrade the global representation over time, which may particularly impact manipulation. Have the authors encountered this problem on the real robot?
3. Why is the SPL metric missing for most baselines in Table 4?

---

> ### Author Rebuttal · Authors · 2026-03-31
>
> We sincerely thank the reviewer for the constructive feedback. For clarity, responses are labeled as W (Weakness), MQ (Minor Question), KQ (Key Question), or L (Limitation).
>
> ## (W1&KQ4) 3D feature map
> Our main contribution is the proposed 3D Interaction Chain (3D-IC), which jointly plans multi-stage navigation and manipulation. The 3D feature map is not a standalone contribution, but an enabling component of 3D-IC. To support VLM-based reasoning over 3D spatial information from different viewpoints, the 3D feature map should satisfy:
>
> - Explicit 3D structure. The map should preserve explicit 3D geometry and spatial layout, since 3D-IC needs to reason about object semantics as well as their locations and spatial relations.
> - VLM Token Alignment. The feature should be aligned with VLM visual tokens, so that the VLM can directly reason over visual observations with its pretrained knowledge.
> - View-specific Rendering. The map should support rendering from multiple viewpoints, since interaction feasibility may vary across views due to viewpoint-dependent spatial constraints.
>
> Following the reviewer’s suggestion, we provide the following comparison table with related works, showing that our 3D feature map differs from existing representations and is specifically designed to our 3D-IC.
> We will discuss our 3D feature map in this context in the revision.
>
> |Method|Representation Space|Explicit Geometric Structure|VLM Token Alignment|View-specific Rendering|
> |--|--|--|--|--|
> |OpenScene|CLIP-aligned features|✔|✘|✘|
> |ConceptFusion|CLIP-fused features|✔|✘|✘
> |ClipFields| CLIP-aligned features|✘|✘|✘|
> |OK-Robot|CLIP-fused features|✔|✘|✘|
> |Ours|VLM tokens|✔|✔|✔|
>
> ---
> ## (W2-1&KQ3) Candidate chains generation
> Not all chains satisfying the rules in Lines 249–265 are directly evaluated using Eq. (4).
> In our implementation, we first pre-rank all candidate chains based on (1) progress toward completion and (2) travel cost, corresponding to the first and third terms in Eq. (4). We then retain only the top-K chains, compute $P(w_k)$ for these candidates, and apply the full Eq. (4) to select the final optimal chain.
> We will add more details in the revision, including both clearer descriptions and K-tuning experiments.
> ## (W2-2) How the set is updated online.
> The waypoint candidate set is updated at each decision step by re-extracting interaction waypoint candidates from the latest map.
>
> This is necessary because navigation waypoints for unobserved objects are selected as frontiers on $\mathcal{M}_t^f$, which change as exploration proceeds.
> Therefore, the candidate set is not incrementally accumulated and must be refreshed from the latest map.
>
> Since this process is deterministic and decision steps are triggered periodically, the computation cost for re-extracting is acceptable.
> ___
> ## (W3-1) Details on fine-tuning
> $P(w_k)$ is produced by the VLM via prompting. We froze the vision encoder and fine-tuned all remaining parameters.
> Regarding the details of data and three-stage fine-tuning, please refer to our response to Reviewer 6Zfq (W2&Q2).
> ___
> ## (W3-2&KQ1&KQ2) Ablation on $P(w)$ and fine-tuning
> We ablate the entire $P(w_k)$ term (Line 1) to isolate the contribution of the VLM-based feasibility score beyond the other terms in Eq. (4).
> We also conduct a step-by-step ablation (Line2-5) of the three-stage fine-tuning pipeline (Token interpretation: TI, Token Preference Learning: TP, Trajectory selection: TS).
> The results confirm the effectiveness of both the $P(w_k)$ term and each fine-tuning stage.
> We will include these results in the revision.
>
> |Method|FindObj.|Pick|FindRec.|Place|Avg. SR
> |:-:|:-:|:-:|:-:|:-:|:-:
> |w/o VLM|56.5|50.9|35.1|11.6|38.5|
> |w/o fine-tuning|65.8|61.2|44.5|12.4|46.0|
> |TI|66.3|62.3|46.2|12.6|46.9|
> |TI+TP|67.5|62.7|50.0|15.9|49.0|
> |TI+TP+TS (Ours)|69.1|64.0|55.6|21.9|52.7|
> ___
> ## (W4&MQ3) SPL
> We will add the citation.
> Our contribution is not inventing SPL, but enabling its evaluation in OVMM by providing the OVMM dataset with the annotations (shortest path) required for SPL.
>
> Since prior works lacked this annotations and omitted SPL, we could only evaluate reproducible methods.
> ___
> ## (MQ1) Euclidean distance in Equation 4
> Since $w_k$ includes manipulation waypoints beyond the 2D navigable space $\mathcal{M}\_t^f$, we use the Euclidean distance between 3D waypoint locations for $d(w_{k-1}, w_k)$.
> ___
> ## (MQ2) The impact of long-term localization and mapping errors
> We mitigate localization and mapping errors across several levels. Methodologically, the 3D feature map retains only the highest-confidence features to preserve semantic stability. During real-world deployment, we apply point cloud smoothing to reduce depth noise. At the execution level, our local policy(AnyGrasp) performs an additional perception check before manipulation.
> Together, these measures help maintain stable manipulation performance on the real robot.
>
>
> **Please let us know if you have further questions.**

---

> > ### Author Rebuttal · Reviewer_jvGm · 2026-04-03
> >
> > I would like to thank the authors for their response. The proposed changes would enhance the manuscript. However, some important concerns remain. The paper would benefit from a major rewrite to deemphasize standard components (mainly the 3D feature map) and properly detail how 3D Interaction Chains are generated and updated over time. I also do not find the empirical validation of the 3-step fine-tuning procedure particularly convincing and hope this will be addressed in the final version.
> >
> > ### 3D Feature Map
> >
> > All compared methods could simply swap in VLM aligned features instead of their original features. Moreoever, if "viewpoint-specific rendering" refers to the ray-casting aggregation scheme in the paper, this looks like a relatively standard procedure to render information from a point cloud map. I think the paper would actually benefit from moving this presentation to the Appendix.
> >
> > To be clear, I understand the role of the 3D feature map as a module is important. However, its design does not appear particularly unique, and I believe the paper would benefit from spending more of its real estate on the actual contributions: the 3D Interaction Chain (3D-IC) and how a VLM is fine-tuned to score them.
> >
> > ### Ablation on $P(w)$ and fine-tuning
> >
> > I would like to thank the authors for the additional results. The provided results are a step in the right direction but do no meet the bar a full ablation. The tasks should be tested separately: Trajectory selection (TS) in particular is likely to work well alone due to the alignment with the main task. I also note that the TI step barely moves the needle, leaving the reader wondering if the simpler TS or TP+TS would perform just as well as the proposed TI+TP+TS. Each step represents a substantial amount of data and curation work and they should be justified carefully.

---

> > > ### Author Response · Authors · 2026-04-05
> > >
> > > We thank the reviewer for the helpful feedback and follow-up suggestions.
> > >
> > > ## 3D feature map
> > > We appreciate the reviewer's valuable suggestion on improving the presentation of the paper.
> > >
> > > In the revision, we will place greater emphasis on our core contribution of 3D-IC, including its generation, online update and VLM-based scoring.
> > > At the same time, we will streamline the discussion of the 3D feature map in the main text. The details of its adaptation for 3D-IC, such as geometric structure, VLM token alignment, and view-specific rendering, will be added to the appendix.
> > > We will revise the main text accordingly to focus more clearly on the core contribution.
> > >
> > > ___
> > > ## Ablation on $P(w)$ and fine-tuning
> > > The motivation of our three-stage fine-tuning is to enable the VLM with different capabilities for scoring interaction chains.
> > > Specifically, TI (Token Interpretation) is introduced because the VLM takes view-organized visual tokens, rather than raw image inputs. TI helps the VLM understand the tokenized representation of each waypoint, including its semantics, 3D geometry, and execution-relevant cues.
> > > TP (Token Preference) then builds on this waypoint-level understanding and trains the model to compare waypoints and contrastively select the better one.
> > > Finally, TS (Trajectory Selection) further extends this reasoning from individual waypoints to full interaction chains, which is most directly aligned with the task, i.e., estimating $\sum_{w}P(w)$ in Eq. (4) of the main text.
> > >
> > > To evaluate effectiveness of each fine-tuning stage, we provide a step-by-step ablation in table below.
> > >
> > > |    |      TI      |      TP      |      TS      | FindObj. | Pick | FindRec. | Place | Avg. SR |
> > > | :--: | :----------: | :----------: | :----------: | :------: | :--: | :------: | :---: | :-----: |
> > > |  1   |              |              |              |   65.8   | 61.2 |   44.5   | 12.4  |  46.0   |
> > > |  2   | $\checkmark$ |              |              |   66.3   | 62.3 |   46.2   | 12.6  |  46.9   |
> > > |  3   |              | $\checkmark$ |              |   66.9   | 62.5 |   47.8   | 13.1  |  47.6   |
> > > |  4   |              |              | $\checkmark$ |   67.7   | 62.9 |   51.2   | 17.2  |  49.8   |
> > > |  5   | $\checkmark$ | $\checkmark$ |              |   67.5   | 62.7 |   50.0   | 15.9  |  49.0   |
> > > |  6   | $\checkmark$ |              | $\checkmark$ |   68.3   | 63.2 |   53.8   | 19.8  |  51.3   |
> > > |  7   |              | $\checkmark$ | $\checkmark$ |   68.2   | 63.6 |   52.1   | 18.3  |  50.6   |
> > > |  8   | $\checkmark$ | $\checkmark$ | $\checkmark$ |   69.1   | 64.0 |   55.6   | 21.9  |  52.7   |
> > >
> > > Comparing rows 3 vs. 5, 4 vs. 6, and 7 vs. 8, we find that adding TI consistently improves performance over training without it. This suggests that TI helps the model better understand the tokenized representation of each waypoint, which in turn benefits the later TP and TS stages.
> > > The smaller gain from TI alone (e.g., rows 1 vs. 2) may be because TI mainly improves representation understanding rather than directly optimizing the task-specific objective.
> > >
> > > Comparing rows 1 vs. 3 and 6 vs. 8 shows that TP also consistently improves performance. This indicates that learning to compare candidate waypoints is beneficial (rows 1 vs. 3) and also helps the subsequent TS stage (rows 6 vs. 8).
> > >
> > > Finally, comparing rows 4 vs. 7 vs. 8 shows that, although TS is directly aligned with the final task, adding TI and TP still provides further gains over using TS alone.
> > >
> > > Overall, these results suggest that TI, TP, and TS are complementary and progressively structured: from understanding a single waypoint, to comparing candidate waypoints, to selecting the best full interaction chain.
> > >
> > > We sincerely thank the reviewer for these detailed and insightful comments, which are very helpful for improving our paper. We will incorporate all these discussions into the revision, and we hope that our response further clarifies the reviewer’s concerns.

---

### Official Review · Reviewer_UQmi · 2026-03-09

**Soundness:** 2
**Presentation:** 2
**Significance:** 2
**Originality:** 3
**Overall Recommendation:** 4
**Confidence:** 3

**Summary:**

The paper proposes a framework that integrates multi-stage navigation and manipulation planning. The proposed 3D-IC method maintains a shared 3D feature map for both skills, generates stage-aligned interaction waypoints, and connects them into candidate multi-stage chains. A hierarchical policy evaluates these chains by jointly considering feasibility—using VLM reasoning over waypoint-centric 3D features—and transition cost, selecting the best balance between success probability and path efficiency. The robot executes the next waypoint and replans as new observations become available.

Claims And Evidence:

The paper claims that a 3D feature map capturing information required for both navigation and manipulation, an interaction chain enabling unified multi-stage planning, and a hierarchical policy that jointly reasons about navigation and manipulation while producing heterogeneous action formats lead to improvements in both stage-wise success rates and overall path efficiency.

Methods And Evaluation Criteria:

The 3D-IC method is evaluated in an OVMM simulator and on a real Stretch 3 robot in real-world environments.

Theoretical Claims:

All the claims were proven through experiments.

Experimental Designs Or Analyses:

The 3D-IC method is evaluated in an OVMM simulator and on a real Stretch 3 robot in real-world environments. The results are compared against the OVMM, UniTeam, MoManip VLA, and MoTo methods.

Supplementary Material:

The supplementary material includes a detailed analysis of failure cases, the impact of VLM fine-tuning, and the prompts used in various tasks.

Relation To Broader Scientific Literature:

The related work is well presented and appropriately compared with prior approaches. However, the literature review primarily focuses on supervised learning, reinforcement learning, and zero-shot methods based on LLMs or VLMs. Classical robotics approaches that address many challenges in mobile manipulation are not sufficiently discussed.

**Compliance With Llm Reviewing Policy:**

Affirmed.

**Final Justification:**

I have increased my score, assuming that the final version will incorporate the promised revisions. However, I remain unconvinced about its overall impact and significance.

**Key Questions For Authors:**

The failure case analysis shows some examples of failures, but it remains unclear why exploration or object-reaching failed and whether these issues can be addressed efficiently.

**Limitations:**

The paper focuses primarily on LLM-, VLM-, and RL-based techniques while largely overlooking classical robotic approaches for 3D environment modeling and motion planning, and the relatively low success rates reported in the experiments are insufficiently analyzed, making it unclear whether the limitations stem from the proposed method or from the OVMM benchmark itself.

**Strengths And Weaknesses:**

Strengths

The paper is well-written and addresses an important problem.

The findings are clearly explained.

The results in Table 4 show strong performance, outperforming the state of the art.

Weakness

I list below what I think are weaknesses:

- I appreciate the outcomes of the paper and the experimental verification. However, the paper focuses on LLM, VLM, and RL techniques while largely ignoring classical robotic structures for 3D environment modeling and motion planning, which could significantly improve the success rate of the proposed method. These approaches should at least be mentioned and discussed. For example, cuRobot can generate manipulator trajectories that avoid collisions.

- The results presented in Table 2 show relatively low success rates, which are mainly influenced by object placement. What causes this low value? Is it an issue with the method itself, or with the OVMM benchmark?

- The statement "A common design is to use \Pi_n to navigate the robot to within a distance threshold of the target (e.g., within 1m), and then switch to πm to generate manipulation actions." overlooks the fact that classical full-body motion planning algorithms, using an environment model (e.g., OctoMap), can produce collision-free paths that allow reliable object grasping.

I list below some technical comments:

- "find recptacle" - should be “find receptacle” in Fig. 1

---

> ### Author Rebuttal · Authors · 2026-03-31
>
> We thank the reviewer for the helpful comments and constructive suggestions.Our responses to each concern are provided below. For clarity, responses are labeled as W (Weakness), Q (Question), or L (Limitation).
>
> ## (W1 & L1) Classical robotic approaches
> Our method focuses on jointly planning navigation and manipulation by predicting high-level interaction waypoints, rather than low-level motion planning.
> Therefore, 3D-IC operates at a different level of planning from the classical robotics methods mentioned by the reviewer.
> The interaction waypoints produced by 3D-IC is then passed to a local policy for execution, where classical robotic methods can be used as local policy.
> Therefore, these methods are complementary to our 3D-IC.
>
> We thank the reviewer for pointing out this relevant methods, and will cite and discuss them in the revision.
>
> ___
>
> ## (W2) Low placement success rates
> The relatively low placement success is mainly due to the intrinsic difficulty of the OVMM benchmark.
> First, the benchmark setup masks out wrist control. For fair comparison, we also disable wrist pose adjustment in simulation, which makes placement more challenging because the robot cannot refine the object pose before release.
> In addition, placement success in OVMM is defined strictly: the object must remain stably placed for 200 timesteps after release, rather than simply making contact with the target receptacle.
> Moreover, the benchmark includes many round or cylindrical objects (e.g., potato, tomato, vase), as well as cluttered receptacles or receptacles with limited support area (e.g., stool, chest of drawers). These factors substantially increase the likelihood of slipping, rolling, or unstable placement.
>
> ___
>
> ## (W3) Statement of ''A common design is to use $π_m$ to navigate..., then switch to $π_m$ to generate manipulation actions.''
> Our statement is intended to describe the common design in the OVMM benchmark, rather than to claim that this decomposition is generally required for robot manipulation.
> In OVMM, the robot operates in a large-scale unseen environment, where target objects and receptacles are typically initially outside the field of view.
> Therefore, existing methods typically decompose the task into:
> - Subtask1: Navigating to within a distance threshold of the target (objects and receptacles)
> - Subtask2: Switching to a manipulation policy once the target is visible, while still allowing minimal base motion during manipulation.
>
> One challenge of OVMM lies in Subtask 1: under partial observations, the agent must reason about where targets are likely to appear and which interaction waypoints are feasible before the scene is fully observed.
> Our 3DIC is designed to plan an interaction chain that jointly spans both the Subtask1 and Subtask2.
> By contrast, classical full-body motion planning methods (e.g., OctoMap) mainly focus on generating collision-free whole-body motions in observed or reconstructed local environments, which is more closely related to the Subtask2 in
> OVMM.
>
> We will clarify this benchmark-specific context and add a more precise discussion of the distinction in the revision.
>
> ___
>
> ## (W4) Typo
>
> Thank you for pointing out this typo. We will correct it in the revision.
>
> ___
>
>
> ## (Q1) Exploration and object-reaching failures
>
> Exploration and object-reaching failures mainly arise from three factors.
> (1) Some OVMM scenes are large and spatially complex, and under the limited timesteps, the agent may fail to sufficiently explore all relevant regions before the episode ends, resulting in target search failures.
> (2) Small target objects (e.g., card, keychains) are more likely to be overlooked during exploration due to their low visual saliency and weak observability in cluttered environments.
> (3) In some cases the robot may become stuck in narrow passages or constrained spaces because its base cannot move omnidirectionally, which not only reduces exploration efficiency but also prevents successful reaching.
>
> ___
>
> ## (L2) Analysis of low success rates
> The analysis of navigation failures is provided in Q1, while the analysis of low placement success rates is discussed in W2. Under the same benchmark setting, our method achieves higher success rates than existing methods while also being more efficient.
>
> **Please let us know if you have further questions.**

---

> > ### Author Rebuttal · Reviewer_UQmi · 2026-04-03
> >
> > Thank you for the clear and thorough responses to my comments. I have increased my score, assuming that the final version will incorporate the promised revisions. However, I remain unconvinced about its overall impact and significance.

---

> > > ### Author Response · Authors · 2026-04-06
> > >
> > > We sincerely thank the reviewer for the thoughtful feedback and are grateful for the increase in score.
> > >
> > > We hope that introducing 3D-IC for cross-stage joint planning will offer a new perspective on the OVMM task beyond previous decoupled pipelines and provide a useful direction for future research.

---

### Official Review · Reviewer_EpLc · 2026-03-12

**Soundness:** 3
**Presentation:** 3
**Significance:** 3
**Originality:** 2
**Overall Recommendation:** 4
**Confidence:** 4

**Summary:**

Overall, this article’s general area is open-vocabulary mobile manipulation (OVMM), which requires robots to perform long-horizon tasks that combine navigation and manipulation in previously unseen environments. Overall, this study’s important contribution concerns proposing 3D Interaction Chains (3D-IC), a unified planning framework that jointly optimizes navigation and manipulation across multiple task stages.

The method constructs a shared 3D feature map from RGB-D observations to represent both navigation and manipulation contexts. Based on this map, the system generates candidate interaction waypoints and links them into multi-stage interaction chains corresponding to OVMM stages (object search, grasp, receptacle search, and placement). A hierarchical policy evaluates these chains using feasibility reasoning via a vision-language model and transition cost, selecting the best plan and executing waypoints with specialized local policies.

Experiments in simulation and real-world environments show that 3D-IC improves both task success rates and path efficiency, outperforming prior OVMM methods.

**Compliance With Llm Reviewing Policy:**

Affirmed.

**Key Questions For Authors:**

What is the runtime cost of generating candidate interaction chains and evaluating them with the VLM during planning? How does this scale with the number of waypoints or environment size?

How sensitive is the system to errors in these upstream modules, and have you evaluated robustness under noisy or incomplete observations?

**Limitations:**

No. The paper does not appear to include a dedicated discussion of limitations or potential societal impacts.
But the author could focus on following parts to improve.
Discuss potential failure modes arising from perception errors (e.g., incorrect object detection, inaccurate 3D mapping) and how these might affect safety and reliability in real-world deployments.
Clarify how well the method is expected to generalize to significantly different environments, object categories, or tasks beyond the OVMM benchmark.

**Strengths And Weaknesses:**

The paper identifies an important limitation of existing OVMM systems—navigation and manipulation are typically planned independently, which can lead to navigation endpoints unsuitable for manipulation. The proposed approach directly addresses this issue by jointly optimizing across stages. The proposed 3D Interaction Chains (3D-IC) provide an elegant abstraction that links navigation and manipulation decisions through interaction waypoints and multi-stage chains, enabling globally consistent planning rather than stage-wise heuristics. The use of a shared 3D feature map that fuses map-level context and egocentric perception provides a principled way to bridge the different input modalities required for navigation and manipulation. The paper does not clearly discuss the runtime overhead of constructing interaction chains and querying the VLM during planning, which may become expensive for long-horizon tasks or large environments. The system depends on accurate open-vocabulary detection, segmentation, and 3D mapping. Errors in these upstream modules could significantly degrade planning performance, but robustness to perception errors is not thoroughly analyzed. Experiments focus primarily on the OVMM benchmark and a single robot platform. Additional comparisons on other embodied benchmarks or environments would strengthen claims of generality.

---

> ### Author Rebuttal · Authors · 2026-03-30
>
> We thank the reviewer for the helpful comments and constructive suggestions.Our responses to each concern are provided below. For clarity, responses are labeled as W (Weakness), Q (Question), or L (Limitation).
>
> ## (W1 & Q1) Runtime overhead
> The planning phase in our 3D-IC consists of two main steps: (1) generating candidate interaction chains via deterministic spatial operations (avg. 273 ms), and (2) selecting the optimal chain with the VLM (avg. 4,290 ms).
>
> This overhead is acceptable for two reasons: (1) 3D-IC planning is performed periodically rather than at every step. (2) real-world physical execution (e.g., grasping) itself takes some runtime, and our planning runs in parallel.
> Therefore, the amortized runtime cost per timestep is modest in practice.
>
> Regarding scalability, candidate-chain generation is based on deterministic geometric operations and incurs relatively low overhead. For VLM-based chain selection, we do not evaluate all candidates. Instead in our implement, we first pre-rank the candidate chains and retain only the top-K for VLM reasoning. For more details, please refer to our response to Reviewer jvGM (W2-1&KQ3).
> This design keeps the runtime controllable and prevents it from growing linearly with the number of waypoints or the environment size.
> We will add these runtime details in the revision.
> ___
>
> ## (W2 & Q2 & L2) Upstream errors
> To evaluate robustness to upstream perception errors, we simulate perception errors by randomly dropping or misaligning 2D semantic segmentation masks before 3D projection.
> As shown in the table below, our 3D-IC remains stable under moderate noise perturbations (<0.15).
> We infer this robustness mainly comes from two factors: (1) our 3D feature map is updated based on confidence and aggregated across multiple viewpoints, so perception errors from a single view are mitigated through multi-view updates. (2) the decision-making in 3D-IC is performed by a VLM, whose reasoning is tolerant to small perception errors (i.e., a few inaccurate visual tokens in the input).
> We will include these robustness analysis in the revision.
>
> |NosieRate|FindObj.|Pick|FindRec.|Place|Avg. SR|
> |:-:|:-:|:-:|:-:|:-:|:-:|
> |0|69.1|64.0|55.6|21.9|52.7|
> |0.10|68.7|62.8|54.3|21.0|51.7|
> |0.15|66.3|62.1|53.1|19.3|50.2|
> |0.20|63.2|58.6|49.9|17.8|47.4|
> ___
>
> ## (W3 & L3) Generalization beyond the OVMM benchmark and a single robot platform
>
>
>
> Since 3D-IC predicts high-level interaction chains rather than task-specific actions, it is not tied to a particular robot platform or benchmark. We further analyze and evaluate its generalization ability across different settings below.
>
> (1) **Other benchmarks.**
> We further evaluate our method on the OWMM[1] benchmark in simulation with the Fetch robot.
> The results show that adding 3D-IC consistently improves performance over the OWMM-VLM baseline across all stages.
>
> ||Robot close to Object|Object Picked|Robot close to Goal|Full Task|
> |--|--|--|--|--|
> |OWMM-VLM|84.6%|38.6%|23.5%|21.9%|
> |OWMM-VLM+3D-IC|88.2%|42.7%|26.9%|24.3%|
>
>
> (2) **Other robot.**
> To evaluate applicability across real robots,
> we additionally deploy our method on the Galaxea R1 Pro and conduct 10 real-world trials. R1 Pro is a mobile manipulation platform with a mobile base, dual 7-DoF arms with grippers, and RGB-D cameras mounted on both the head and wrists. Results show that the method transfers to different robotic embodiments.
>
> ||FindObj.|Pick|FindRec.|Place|
> |--|--|--|--|--|
> |Heur.|70%|30%|20%|0%|
> |3D-IC|80%|50%|50%|30%|
>
> (3) **Other task.**
> We further evaluate our method on MON (multi-object navigation tasks)[2].
> The results show that the proposed interaction-chain formulation is also applicable with multi-stage navigation task.
>
> ||2-MON|3-MON|4-MON|
> |--|--|--|--|
> |SGM|25.5|11.9|8.6|
> |GDWO|32.4|19.1|14.3|
> |3D-IC|35.8|23.4|17.4|
>
> Overall, these results suggest that our method generalizes beyond different benchmark, robot embodiment and task.
>
>
> ___
>
> ## (L1) Lack of limitations
> In the final version, we will add a dedicated section discussing our limitations and societal impacts.
>
> ___
>
> ## Reference
> [1] OWMM-Agent: Open World Mobile Manipulation With Multi-modal Agentic Data Synthesis, NeurIPS 2025
> [2] Goal-oriented Dynamic Weight Optimization for
> Multi-Object Navigation, TPAMI 2026
>
> **Please let us know if you have further questions.**

---

> > ### Author Rebuttal · Reviewer_EpLc · 2026-04-03
> >
> > Thanks author for rebuttal.
> > My question is fully resolved but for icml I hope there is more method contribution. I like this work. I remain my score.

---

> > > ### Author Response · Authors · 2026-04-06
> > >
> > > We sincerely thank the reviewer for the positive feedback and are especially encouraged to hear that the reviewer likes our work.
> > >
> > > Regarding the methodological contribution, our proposed 3D-IC introduces a unified abstraction that links navigation and manipulation via an interaction chain, enabling joint planning within a single framework. We believe this joint planning framework represents a meaningful methodological step for OVMM tasks.

---

### Official Review · Reviewer_6Zfq · 2026-03-19

**Soundness:** 3
**Presentation:** 3
**Significance:** 3
**Originality:** 3
**Overall Recommendation:** 4
**Confidence:** 4

**Summary:**

This paper focuses on open-vocabulary mobile manipulation (OVMM), where a robot operates in a new environment and needs to find a target object, pick it up, move to a target receptacle, and place it there. The authors point out that many existing methods treat navigation and manipulation as separate problems, which may work locally but often leads to suboptimal decisions for long-horizon tasks.

To solve this, they propose 3D Interaction Chains (3D-IC), a hierarchical framework that plans the whole task more globally. The method first builds a shared 3D feature map from RGB-D observations, then generates stage-specific interaction waypoints for different sub-tasks, including object search, grasping, receptacle search, and placement. These waypoints are connected into candidate multi-stage action chains. A VLM-based high-level policy then scores these chains by considering both feasibility and path cost, while low-level navigation and manipulation modules execute the next chosen waypoint.

Experiments on the OVMM benchmark show that 3D-IC outperforms heuristic methods, RL-based approaches, and recent mobile manipulation baselines. Real-world experiments on a Stretch 3 robot further demonstrate its advantages, particularly in more challenging cross-room tasks.

**Compliance With Llm Reviewing Policy:**

Affirmed.

**Final Justification:**

This paper tackles a relevant and challenging problem in OVMM and proposes a clear system-level framework for joint planning across navigation and manipulation stages. The method is reasonably motivated and supported by strong results on both simulation benchmarks and real-robot experiments. Although the technical novelty is more in the overall formulation and integration than in a fundamentally new algorithm, and some details could still be explained more clearly, I find the contribution valuable and sufficiently solid for acceptance.

**Key Questions For Authors:**

1. The paper introduces a waypoint feasibility probability P(w), but it is unclear how this scalar probability is obtained in practice from the model's generative outputs. Is it derived from token probabilities, post-hoc scoring, or an additional calibration step?
2.How sensitive is performance to candidate waypoint coverage? In particular, are failures primarily due to poor high-level scoring or due to missing good grasp/placement candidates during waypoint proposal generation?
3. Can the authors provide more detail on the construction of the 120K training samples, including how much manual refinement is required, how failure attributions are produced, and how samples are distributed across the different training tasks?

**Limitations:**

No.
Although it includes real-robot experiments,  runtime efficiency is not reported clearly enough: the system relies on online 3D map updates, candidate waypoint/chain construction, and high-level decision making with a large VLM, yet key metrics such as latency, FPS, and real-time suitability are not provided.

**Strengths And Weaknesses:**

Strengths:

The paper studies an important problem in embodied AI and is well aligned with the OVMM setting. The motivation is clear, and the method also achieves state-of-the-art performance on the benchmark.

The proposed method is conceptually clear and well structured. The shared 3D feature map, stage-specific waypoint generation, candidate chain construction, and the trade-off between feasibility and path cost together form a fairly complete planning framework.

Weaknesses:

The main contribution is more about planning formulation and system-level integration than about introducing a fundamentally new algorithmic component. Many of the building blocks, such as 3D mapping, frontier exploration, and off-the-shelf execution modules, are already known.

The training data construction is not described clearly enough. The paper mentions 120K instruction-following samples collected through teacher-model distillation and manual refinement, but it does not give enough detail about the annotation effort, data composition, or how errors are attributed.

The paper introduces a waypoint feasibility probability P(w), but it is still unclear how this scalar is actually derived from the model’s generative outputs in practice.

---

> ### Author Rebuttal · Authors · 2026-03-31
>
> We sincerely thank the reviewer for the constructive feedback. For clarity, responses are labeled as W (Weakness), Q (Question), or L (Limitation).
> ## (W1) Contribution
> Our core contribution is the proposed 3D Interaction Chain (3D-IC), which unifies navigation and manipulation within a common abstraction and enables joint planning across multiple navigation and manipulation stages. To achieve this, the key blocks of 3D-IC (interaction-chain formulation, interaction-waypoint sampling, and optimal chain selection) are newly designed algorithmic components.
>
> The blocks mentioned by the reviewer (e.g., 3D mapping, frontier exploration, and off-the-shelf execution) are used to support 3D-IC rather than being the main contributions themselves. To the best of our knowledge, such joint planning across multiple navigation and manipulation stages has not been explored in prior OVMM. Moreover, our contribution is orthogonal to existing blocks and can complement them to further improve OVMM performance.
> ___
> ## (W2 & Q2) Training data and fine-tuning details
> Our fine-tuning objective is to enable the VLM to reason about waypoint feasibility. The training data construction includes data collection, automated annotation, and manual post-verification.
>
> For data collection, we traverse training scenes to construct 3D feature maps, from which interaction waypoints, 3D tokens, and valid trajectories are collected.
> Then we execute actions on sampled waypoints to gather simulator feedback and post-execution RGB images. For navigation, feedback indicates whether the agent moves closer to the goal. For manipulation, we acquire detailed feedback from the simulator, including object-receptacle contact, collisions, and position errors. These physical cues are provided to the annotator VLM to attribute the failure reason (e.g.,the object was placed near the receptacle boundary and eventually slipped because the placement orientation was not aligned with the receptacle edge).
>
> For automated annotation, the annotator VLM (teacher-model in main text) generates annotation from the collected data.
> - Token interpretation(30K examples). Given a waypoint, the annotator VLM generates a concise description, covering (1) semantic role, (2) local 3D geometry, and (3) execution-relevant cues(e.g. reachability, grasp stability, and placement suitability).
> - Token preference learning(30K examples). Given two candidate waypoints from the same instance and stage, the annotator VLM provides a concise attribution and scores, based on the simulator feedback. The attribution captures relative trade-offs (e.g., accessibility vs. stability, collision risk vs. support quality), encouraging decision-oriented comparison rather than isolated waypoint description.
> - Trajectory selection(60K examples). Given multiple candidate trajectories from the same task, the annotator VLM analyzes the simulator feedback for each waypoint along each trajectory and provides corresponding reason and scores. These annotations are used to further facilitate the understanding and selection of entire trajectories.
>
> For manual post-verification, the final refinement takes 120 person-hours (5 checkers over 3 days) and primarily filters out low-quality semantic annotations.
>
> Finally, we fine-tune the model (with the vision encoder frozen) sequentially in three stages. Through fine-tuning, 3D-IC learns to reason about waypoint and trajectory feasibility based on the 3D tokens.
> ___
>
> ## (W3 & Q1-1) Derivation of $P(w)$
> During inference, the VLM  evaluates candidate trajectories and generate numerical feasibility scores in a predefined format, from which we parse $\sum_w P(w)$.
> ___
>
> ## (Q1-2) Sensitivity to candidate waypoint coverage and sources of failure.
> To evaluate the impact of interaction waypoint coverage, we compare performance under different waypoint sampling intervals, where 0.3 means sampling every 0.3 m.
> Experimental results (shown in the table below) indicate that performance no longer improves when the sampling interval is below 0.3 m, suggesting that good grasp/placement candidates are already sufficiently covered at this density.
> This also indicates that our method is relatively insensitive to the sampling interval within a reasonable range (0.10–0.30 m).
> Regarding failures, most are related to the OVMM benchmark setting rather than insufficient waypoint coverage. For a more detailed failure analysis, please refer to our response to Reviewer UQmi (W2).
>
> |SamplingInterval|FindObj.|Pick|FindRec.|Place|Avg. SR|
> |:-:|:-:|:-:|:-:|:-:|:-:|
> |0.10|68.9|63.8|54.8|20.6|52.0|
> |0.20|69.1|64.2|55.1|21.8|52.6|
> |0.30|69.1|64.0|55.6|21.9|52.7|
> |0.40|68.8|63.0|54.4|20.9|51.8|
> |0.60|66.9|61.3|53.6|18.7|50.1|
> |0.80|65.3|59.1|52.1|16.5|48.3|
> ___
> ## (L1) Runtime efficiency
> We will include this analysis in the revision.
> For more details, please refer to our response to Reviewer EpLc (W1&Q1).
>
> **Please let us know if you have further questions.**

---

> > ### Author Rebuttal · Reviewer_6Zfq · 2026-04-03
> >
> > Thank you for the detailed rebuttal. The response clarifies several of my original concerns, especially the construction of the 120K training data, the role of simulator feedback and manual post-verification, and the sensitivity to waypoint sampling coverage. The explanation of the contribution also better positions 3D-IC as a joint planning abstraction over multiple navigation and manipulation stages, rather than merely a loose system integration.
> >
> > However I still think the paper would benefit from a clearer description in the main text of how the waypoint feasibility score / P(w) is concretely produced from the VLM outputs during inference, ideally with a precise example of the scoring/parsing procedure.
> >
> > Overall, my concerns are partially resolved. The remaining issues seem addressable through clarification and additional reporting rather than indicating a fundamental flaw in the approach, so I keep my overall assessment unchanged.

---

> > > ### Author Response · Authors · 2026-04-03
> > >
> > > We thank the reviewer for the follow-up and suggestions.
> > > We further clarify how the feasibility score $P(w)$ is produced from the VLM during inference.
> > >
> > > Specifically, we provide a prompt template for VLM inference in **[infer-prompt-template URL](https://anonymous.4open.science/r/anonymous-resource-BD28/3D-IC/infer-prompt-template.txt)**. Under this prompt, the VLM is instructed to first assess the feasibility score $P(w)$ for each interaction waypoint, then score each interaction chain accordingly, and finally output the reasoning, together with the score of each chain (i.e., $\sum_w P(w)$ in Eq. 4 of the main text).
> > >
> > > To make this process more concrete, we also provide an example of the parsing procedure in **[infer-example URL](https://anonymous.4open.science/r/anonymous-resource-BD28/3D-IC/infer-example.md)**, including the prompt input to the VLM and the corresponding format output. Note that the actual input also includes the 3D visual tokens rendered from the 3D feature map. We denote them as `<3D tokens>` for simplicity in the example.
> > >
> > > We appreciate the reviewer’s valuable suggestion and will include both the inference prompt template and the parsing example in the revision.

---

### Decision · Program_Chairs · 2026-04-30

**Decision:**

Accept (regular)

**Comment:**

The authors propose 3d interaction chains (3d-ic), a framework for joint navigation and manipulation planning. They use a shared 3d feature map to generate interaction waypoints, then score them using a VLM. This allows for improved performance on open vocabulary mobile manipulation tasks.

The method itself is well motivated, and supported by strong results in simulation and on real robots. It looks at a problem that's often under explored -- how to jointly reason over both base and arm motions.

The work could be better explained in some areas, particularly in the construction of their training data and how the feasibility score is generated. It's main weakness is a lack of algorithmic depth, but it shows very strong engineering and interesting results.